# Translation, Adaptation and Validation of the Five-Word Test (Test Delle 5 Parole, T5P) in an Italian Sample: A Rapid Screening for the Assessment of Memory Impairment

**DOI:** 10.3390/geriatrics7020049

**Published:** 2022-04-15

**Authors:** Elisabetta Farina, Francesca Borgnis, Susanna Scioli, Alessia d’Arma, Alessandra D’Amico, Francesca Lea Saibene

**Affiliations:** IRCCS Fondazione Don Carlo Gnocchi ONLUS, 20148 Milan, Italy; efarina@dongnocchi.it (E.F.); sscioli@dongnocchi.it (S.S.); adarma@dongnocchi.it (A.d.); aldamico@dongnocchi.it (A.D.); fsaibene@dongnocchi.it (F.L.S.)

**Keywords:** Alzheimer’s disease, mild cognitive impairment, functional memory disorders, five-word test, Test delle 5 Parole-T5P, neuropsychological assessment, hippocampal memory, adaptation, validation, screening memory test

## Abstract

Discriminating between Alzheimer’s disease (AD), mild cognitive impairment (MCI) and functional memory disorders (FMD) can sometimes be difficult because amnesic syndrome can be confused with memory dysfunction due to aging and/or depression. This work aims to translate, adapt and validate the five-word test (FWT) in an Italian sample (Test delle 5 Parole, T5P). FWT is a screening memory task specifically built to discriminate the hippocampal memory deficit—typical of AD and amnesic MCI—from those related to functional disorders. We involved a large number of subjects (*n* = 264): 60 with AD, 80 with MCI, 46 with FMD, and 78 healthy controls (CTRL). Two traditional screening tests and the T5P were administered to all participants. Results showed an excellent convergent validity of the T5P (*p* < 0.05). Moreover, the T5P appeared as an effective test able to discriminate between the four conditions, except for FMD and CTRL. In particular, a total score of ≤9 allowed for accurately discriminating subjects with AD + MCI compared to participants with FMD + CTRL with high sensitivity (83.87%) and specificity (76.43%). Overall, the T5P is a simple, rapid and sensible test of the hippocampal memory that is recommended in clinical practice to screen persons with subjective/probable amnesic difficulties and to discriminate between patients with AD or MCI and CTRL.

## 1. Introduction

In recent years, there has been a growing interest in the early detection of Alzheimer’s disease (AD) for prompt, short-term action. Identifying patients in the prodromal phase could be pivotal for inclusion in clinical trials and new possible disease-modifying therapy (e.g., Aducanumab-avwa) [1]. AD is the most common form of dementia [2], currently affecting approximately 600,000 people in Italy. The rate of development of AD is heightened in individuals with the amnestic form of mild cognitive impairment (MCI) [3,4]: individuals with MCI convert to AD at a rate of nearly 10% to 25% per year, in comparison to approximately 1–2% per year in normal aging [5,6,7], making it imperative to generate effective methods for identifying individuals with MCI. Therefore, the need to early detect memory disorders related to AD and amnesic MCI primarily requires developing tools and instruments that are simple, rapid, sensitive, and specific to memory deficits [8,9]. If memory disorder is a cardinal sign of AD, it is also true that amnestic disorders can be linked to different pathological conditions, such as depressive and anxious disorders, sleep disorders, internist pathologies, other types of dementia (e.g., vascular dementia) or neurodegenerative diseases (e.g., frontotemporal dementia, normotensive hydrocephalus) and the use of anticholinergic drugs and benzodiazepines [10]. Recently, the condition, known as “subjective memory complaints”, has received increasing interest since many epidemiological studies have shown that this category of patients is associated with a higher risk of developing MCI or AD [11].

Discriminating the spectrum between Alzheimer’s disease (AD), mild cognitive impairment (MCI), and functional amnestic disorders (subjective and due to depression) can sometimes be difficult because the hippocampal amnesic syndrome typical of AD can be confused with memory dysfunction due to age, depression or no-AD pathology [8,10]. The Free and Cued Semantic Reminding Test (FCSRT; [12]) is a memory test specifically built to discriminate the hippocampal memory deficit—typical of the AD—from those related to functional disorders or other no-AD pathology (e.g., frontotemporal dementia). A simplified version of this test would be useful as screening if available to general practitioners and specialists.

In an attempt to achieve these objectives, in this study, we have translated into Italian and standardized the ‘5 Words Test by Dubois’ (the five-word test, FWT; in Italian: Test delle 5 Parole, T5P) [13], a serial verbal memory test with semantic cueing. The test’s construction aims to identify patients that exhibit memory disorders, possibly due to hippocampal dysfunction.

The aim of this work is to replicate the results of the original paper by Dubois (2002) in an Italian sample, following the same methodology [13]. Specifically, the work consisted of the translation, adaptation and validation of T5P in an Italian sample for the differential diagnosis between AD, amnesic MCI and functional memory disorders (FMD), comparing it with two tools usually used in the clinical field for a first cognitive screening: The Mini-mental State Examination (MMSE, [14,15]) and the clock drawing test (CDT) [16]. We expect that T5P will help clinicians in discriminating between persons with AD, FMD and MCI. Moreover, we expect that T5P will not distinguish between healthy controls and functional memory disorders that involve persons with subjective amnestic impairments and amnestic disorders due to depressive syndrome (in which persons should be able to achieve normal performance with the cue).

## 2. Materials and Methods

The five-word test has been translated into the Italian language by a native speaker according to the methodology of translation and back-translation technique [17] according to Dubois’ instructions and tracing the original construction presented in Dubois’ paper [13].

We kept the original five semantic categories of FWT (buildings, insects, drinks, vehicles and kitchen utensils) and selected, for each, one of the most prototypical words in the Italian language (not among the first three) [18]. A preliminary list of 5 words (museum, *dragonfly*, orangeade, truck, funnel) was administered to a pilot sample of 20 elderly subjects, observing that most of them were unable to associate the word dragonfly with the insect category. Therefore, we replaced this word with ‘firefly’, obtaining greater consensus and recognition. Overall, the final five-word list included: museum *(in Italian museo)*, semantic cue: building; *firefly (in Italian: lucciola)*, semantic cue: insect; orangeade *(in Italian: aranciata)*, semantic cue: drink; truck *(in Italian: camion)*, semantic cue: vehicles; funnel *(in Italian: imbuto)*, semantic cue: kitchen utensil.

Following the original work, instructions for administering the test were translated and drafted. Detailed instructions to administer the test (in English and Italian), T5P stimuli and response forms are in the Appendix A. In summary, the administration of T5P involves four main phases: (1) coding/learning of five words, (2) immediate recall, (3) interference task (i.e., the CDT), and (4) delayed recall. Therefore, the test allows collecting three scores: (1) learning/immediate recall (range 0–5), (2) delay recall (range 0–5), and (3) total score (the sum of the other two scores, range 0–10).

The subjects aged ≥ 60 and with ≥3 years of education were recruited by the Cognitive Disorders and Dementia Center and the Neurology Clinic of IRCCS, Santa Maria Nascente (Fondazione Don Carlo Gnocchi ONLUS, Milan, Italy).

We recruited persons with AD diagnosis, MCI and functional memory disorders. Diagnoses were made according to recognized clinical criteria [4,19] and to a standardized protocol that involved clinical evaluation (including autonomy in daily life) by the neurologist, magnetic resonance imaging (MRI), or computed tomography (CT), neuropsychological tests and blood tests. When appropriated, the neurologist has prescribed other second-level diagnostic tests (e.g., FDG PET, amyloid PET, EEG, CSF examination, etc.) to detect any clinical biomarkers. Moreover, all participants underwent an evaluation of depression through the use of the Hamilton depression scale [20]. Functional memory disorders (FMD) involved persons with subjective amnestic impairments and amnestic disorders due to depressive syndrome (Hamilton score > 14). Exclusion criteria involved: (1) other dementia; (2) systemic or motor disease; (3) other psychiatric disorders other than depression.

The control subjects were recruited among volunteers, family members and people participating in the public meeting. The sample was then enlisted (*n* Total = 264 subjects): the MMSE, T5P and CDT were administered to each subject upon subscription of an informed consent. The study was performed in accordance with the principles of the Helsinki Declaration and by previous approval from the IRCSS Fondazione Don Carlo Gnocchi ONLUS.

### Statistical Analysis

Descriptive statistics included the frequencies, median and interquartile range (IQR) for categorical variables and the mean and standard deviation (SD) for continuous measures. The normality of data distribution was assessed using the Kolmogorov–Smirnov test. ANOVA between-group comparisons (post hoc: Bonferroni) were conducted to assess significant age differences. Moreover, the Kruskal–Wallis and Chi-square tests evaluated possible differences in education (variable not normally distributed) and gender. The ANCOVA between-subject (covariate: age) was conducted to assess any differences in neuropsychological tests due to diagnosis. Moreover, a partial correlation was applied to compare the scores of the neuropsychological tests. Finally, ROC curves evaluated all the tests’ specificity and sensitivity. All statistical analyses were performed using Jamovi 1.6.7.

## 3. Results

### 3.1. Participants

Table 1 presents the sample’s demographic characteristics (*n* = 264) divided between the four groups.

Firstly, the ANOVA between-subject revealed statistically significant differences in ages in the four groups (F (3260) = 6.86, *p* < 0.001). However, the post hoc test (Bonferroni) demonstrated only a difference between AD and FMD (*p* < 0.001). For this reason, the following analyses were conducted, inserting the variable “age” as the covariate. Moreover, the Kruskal–Wallis test did not reveal statistically significant differences in education between the four groups (*p* > 0.05). Finally, the comparison of sex between groups using the Chi-square test did not reveal statistically significant differences (*p* > 0.05).

Interestingly, the sample size calculation performed with G power software (see Table 2) showed that our total sample allowed us to obtain an excellent statistical power = 0.999.

### 3.2. Neuropsychological Assessment

Table 3 offers an overview of the scores obtained from the neuropsychological evaluation divided between all four groups.

When controlling for age, the ANCOVA between-subject (covariate: age) revealed statistically significant differences in the neuropsychological tests (*p* < *0*.001): MMSE (F (3,259) = 135.11, *p* < 0.001), T5P (F (3,259) = 73.06, *p* < 0.001, η^2^ = 0.456) and CDT (F (3,259) = 28.79, *p* < 0.001).

Figure 1 reveals that the post hoc test (Bonferroni) showed significant differences in the MMSE score (*p* < 0.001) between ADvsMCI, ADvsFMD, ADvsCTRL, MCIvsFMD, MCIvsCTRL, but not CTRLvsFMD (*p* > 0.05).

Similarly, the post hoc test (Bonferroni) conducted on the T5P score showed significant differences (*p* < 0.001) between ADvsMCI, ADvsFMD, ADvsCTRL, MCIvsFMD, MCIvsCTRL, but not CTRLvsFMD (*p* > 0.05) (Figure 2). Interestingly, data showed that the variable “age” did not significantly affect the results (*p* = 0.105).

Finally, for the CDT score, the post hoc test (Bonferroni) shows significant differences (*p* < 0.001) between ADvsMCI, ADvsFMD, ADvsCTRL, MCIvsCTRL, but not MCIvsFMD (*p* = 0.111) and CTRLvsFMD (*p* = 0.238).

Interestingly, as expected, a statistical analysis showed a positive linear correlation between age (covariate) and all neuropsychological tests. Moreover, a positive correlation appeared between the two traditional neuropsychological tests and education. Furthermore, a correlation appears between the CDT and sex (*p* < 0.05). Finally, the T5P (total score) does not show a correlation with other demographic characteristics.

### 3.3. Correlation between Neuropsychological Tests

Table 4 shows the correlations between all three neuropsychological tests. Partial correlation showed a positive linear correlation between T5P (total score) and, respectively, the MMSE total score (*p* < 0.001) and the CDT (*p* < 0.001) (Table 4). Moreover, the T5P Free Delayed Recall significantly correlates with the MMSE 3 Words Recall (*p* < 0.001).

### 3.4. Classification of Healthy Controls or Clinical Group

The performance of the classifiers was evaluated by carrying out a relative operating characteristic (ROC) analysis. The area under the ROC curve (AUC) provides a single measure of overall prediction accuracy. Specifically, ROC curves investigated the diagnostic accuracy of the T5P: a total score of ≤9 could accurately discriminate the PAT group (AD + MCI) from the HC Group (FMD + CTRL) with high sensitivity (83.87%) and specificity (76.43%) (AUC = 0.845—excellent accuracy value) (Figure 3).

Table 5 reveals the ability of a T5P total score ≤ 9 for discriminating between different groups with good sensitivity and specificity (except for MCIvsFMD/CTRL) (Table 4).

Further frequency analysis shows that 96.7% of the AD group and 61.3% of the MCI group show scores ≤ 9. On the contrary, 87% of the FMD group and 82.1% of the CTRL group show a total score = 10.

Interestingly, a T5P total score of ≤ 9 can discriminate between FMD and MCI with slightly better sensitivity than the MMSE total score (Figure 4). Specifically, although the overall prediction accuracy (AUC) was statistically better for the MMSE (AUC = 0.849, CI = 0.781–0.918) than T5P (AUC = 0.760, CI = 0.678–0.842) (DeLong test—*p* < 0.05), the T5P shows a slightly better sensitivity (86.96% vs. 84.78%) and a similar specificity (61.25% vs. 68.75%). Regarding the CDT, the T5P obtains better overall prediction accuracy (DeLong test—*p* < 0.05) and sensitivity (86.96% vs. 60.87%) with similar specificity (61.25% vs. 63.75%).

Finally, a total score of ≤ 8 could accurately discriminate the AD group from the MCI group: AUC = 0.778 (acceptable) with high specificity (86.67%) and mild sensitivity (56.25%).

In line with Dubois’ original paper (2002) and following his suggestions, a further analysis was conducted to observe if the combination of cut-off = 9 and intrusions (i.e., a T5P total score of ≤9 and/or one or more intrusions) could provide more sensitive discrimination between groups. Results show that a T5P total score of ≤9 and/or the presence of intrusions could accurately discriminate the pathologic group (AD and MCI) and HC (FMD and CTRL) (AUC = 0.796—acceptable) with better specificity (79.3%) but worse sensitivity (79.8) than just considering the cut-off). Specifically, demographic analysis shows that 96.7% of the AD group and 66.3% of the MCI group show pathological scores. On the contrary, 78.3% of the FMD group and 80.8% of the CTRL group show a total score = 10 and the absence of intrusions.

## 4. Discussion

This work fits the growing interest in early detection, and consequent timely treatment of AD and MCI, which could be a prodromal phase of AD [21]: individuals with MCI convert to AD at a rate of nearly 10% to 25% per year, in comparison to approximately 1–2% per year in normal aging [5,6,7]. To this aim, developing a simple, rapid and sensitive tool that can detect early memory deficits appears critical [8,9]. Early identification of memory deficits suggestive of AD onset could allow for addressing people to further appropriate and specific exams to confirm the diagnosis. In this way, identifying patients in the prodromal phase could be pivotal for inclusion in clinical trials and new possible disease-modifying therapies that could enable individuals to live long and productive lives free of dementia caused by AD [22].

In this framework, we proposed the translation, adaptation and validation of the T5P in an Italian sample as a quick and sensible test of hippocampal memory, able to discriminate between patients with AD, subjects with MCI, and healthy controls. Firstly, we have translated the original version of the five-word test into the Italian language [13]. Unlike previous studies [23], we adapted the test to the Italian language during the translation process from French to Italian, demonstrating that a literal translation was not satisfactory in a small sample. Then, we validated this adapted version in a large sample (*n* = 264) consisting of 60 patients with AD, 80 subjects with MCI, 46 participants with FMD, and 78 healthy controls, matched for education and gender. To our knowledge, this study is also the only one that carried out a calculation of the sample size: we demonstrated that the sample we used allows for obtaining an excellent statistical power.

All participants underwent a short evaluation that involved two traditional standardized screening tests, the MMSE and CDT, along with our version of T5P. Data showed an excellent convergent validity of T5P as demonstrated by the significant correlation between this test and the other two standardized screening neuropsychological tools. Therefore, the T5P can be considered a promising screening test to detect memory deficits. Furthermore, the T5P also appeared as an effective test able to discriminate between the four conditions (AD, MCI, FMD and normality), except for FMD and the normal controls—as we expected, no differences appeared between CTRL and functional memory disorders in all three tests. Persons with FMD obtained high performance comparable to healthy controls since they presented subjective amnestic disorders or memory deficits due to depressive syndrome (subjects were able to achieve normal performance with the cue) [10].

Further analyses demonstrated an excellent diagnostic accuracy of T5P. Specifically, a total score of ≤9 allows accurately discriminating subjects with AD and MCI compared to participants with FMD and healthy controls with high sensitivity and specificity. According to the literature [24], an accuracy value between 0.8 and 0.9 can be considered an excellent value.

Our results confirm the content that the T5P can be useful to detect memory dysfunction in people with AD in accordance with previous works [13,23]. In particular, our results reproduced those of the original work of Dubois and colleagues that showed the ability of the T5P total score of ≤9 to distinguish patients with Alzheimer’s from subjects complaining of FMD, with excellent sensitivity and specificity. As in Dubois’ sample, the performance of patients with AD appeared to be characterized by a reduced free recall and partial indexing help, responsible for a total score of <10. This pattern is characteristic of the “hippocampal” amnesic syndrome observed during AD, even at the beginning of the disease [25]. In fact, normal elderly subjects or those who complain of memory problems without brain pathology obtain a normal performance, immediately or with the help of the cues, thus reaching the total score of 10.

In addition to Dubois’ type of population, in our work, we also selected subjects with MCI because this clinical condition could be considered the prodromal phase of AD [21]: the conversion rate of MCI to AD is estimated to be between 10–25% per year [5,6,7]. Moreover, a recent systematic review has shown that verbal memory tests are excellent to predict MCI individuals who will convert to dementia and should be a critical component of any evaluation intended to identify AD at the predementia stage [26]. Our results confirm that the T5P can also be useful to detect memory dysfunction in people with MCI in accordance with previous work [23]. Differently than in that work, we also tested subjects with functional memory disorders to evaluate the ability of the T5P to distinguish between MCI and persons with subjective amnestic disorders and amnestic difficulties linked to depressive mood. Data also showed that our version of the T5P allows us to accurately discriminate the AD group from the MCI one with mild sensitivity but high specificity. The ability of the T5P to distinguish between AD, MCI and functional memory disorders appears crucial in the differential diagnosis of memory disturbances because the normal age-related reduction of memory performances can be exacerbated by different pathological conditions such as depression [27]. Thus, AD amnesic syndrome can be sometimes confused with functional memory dysfunction due to depression [28].

Interestingly, compared to the other two neuropsychological tests, the T5P total score of ≤9 is able to discriminate between FMD and MCI with better sensitivity and similar specificity than the MMSE and CDT (compared to the CDT, the T5P also obtained a better overall prediction accuracy). This result appears promising because the T5P is a free test that general practitioners could use for a preliminary screening before sending patients to a specialist. In line with Dubois’ suggestions that noted a trend of patients with AD to produce intrusions, often prototypical, we studied whether the combination of the cut-off score and the presence of intrusions provided more sensitive discrimination between groups. However, this combination did not offer advantages compared to the use of the cut-off only. On the contrary, it discriminated between the pathologic groups and normal controls with worse overall prediction accuracy and sensitivity.

Overall, the T5P is a simple, rapid (about 5 min, including an interval between learning and recall) and sensible test of hippocampal memory that is recommended in routine clinical practice to screen persons with functional amnesic difficulties and discriminate between patients with AD or MCI and healthy controls (including persons with subjective or due to depressive mood memory problems). In other words, the T5P can be a valuable solution to screen patients to individuate a probable hippocampal amnesic syndrome. Importantly, the T5P is a useful test for screening memory disorders, not for a clinical and complete diagnosis. When considering the increasing frequency of memory disorders in the general population (memory is the most complained problem by patients), this simple but effective tool could help general practitioners to discriminate between degenerative hippocampal memory deficits and “functional” disturbances, providing a filtering and guiding role (e.g., to prescribe an extended neuropsychological evaluation). Compared to the T5P, the CDT is unable to evaluate the anterograde memory so the performance may be normal in patients with MCI. In addition, the MMSE is a screening test that allows for assessing memory impairment but without cue that could enable to distinguish between a pathological sample and persons with functional memory disorders.

A limitation of this work is regarded as not considering all types of dementia (e.g., vascular dementia) or neurodegenerative disorders (e.g., frontotemporal dementia). However, we have focused on the most common dementia [2], above all, in the Italian Cognitive Disorders and Dementia Center. Moreover, our primary aim was to replicate the Dubois’ study, adapting the test to the Italian population. Further studies could be conducted to expand the clinical sample involved.

## Figures and Tables

**Figure 1 geriatrics-07-00049-f001:**
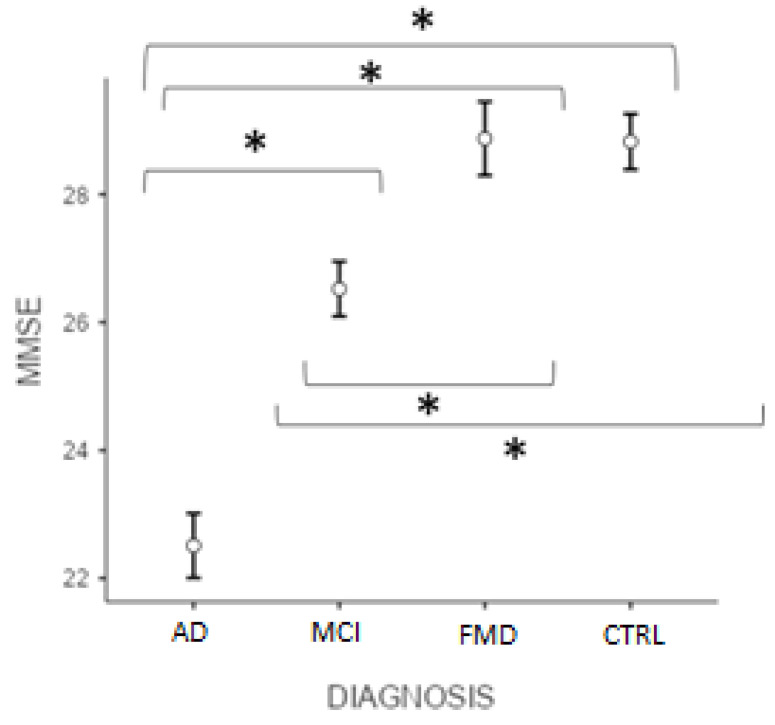
Differences between groups at MMSE total score. (* significant different, *p* < 0.001). AD = Alzheimer’s Disease; MCI = mild cognitive impairment; FMD = functional memory disorders; CTRL = controls.

**Figure 2 geriatrics-07-00049-f002:**
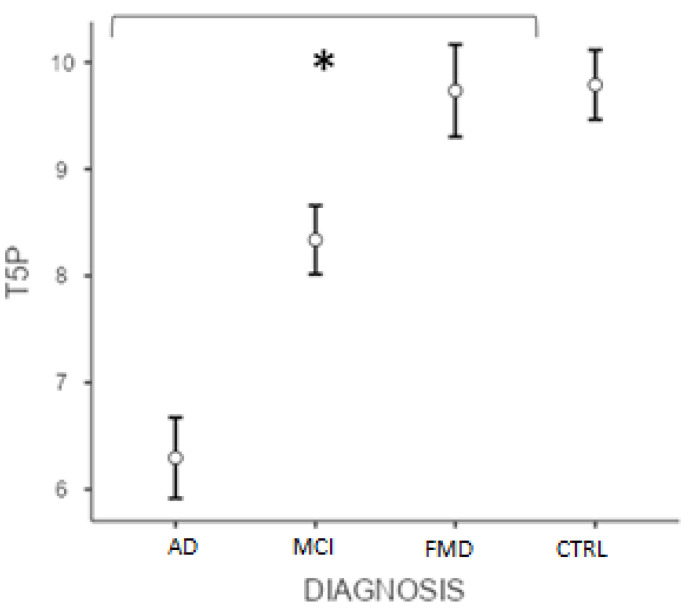
Differences between groups at T5P total score. (* significant different, *p* < 0.001). AD = Alzheimer’s Disease; MCI = mild cognitive impairment; FMD = functional memory disorders; CTRL = controls.

**Figure 3 geriatrics-07-00049-f003:**
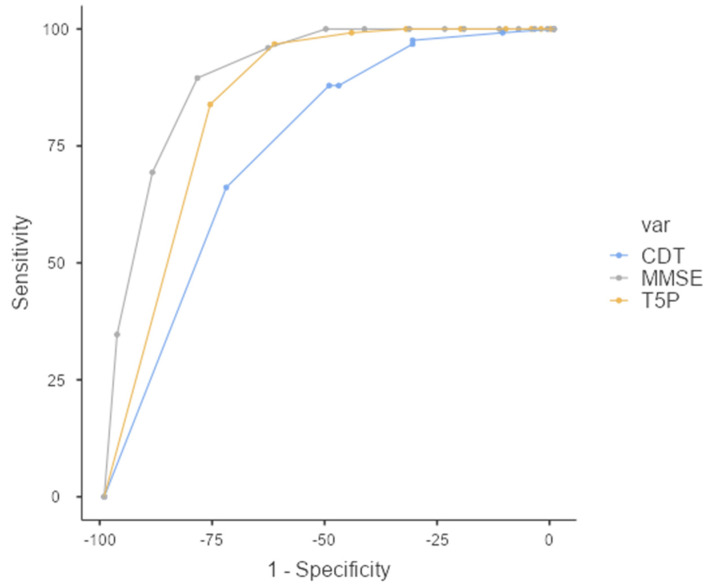
ROC Curve—T5P total score ≤ 9 discriminates between PAT group and HC group. MMSE: Mini-Mental State Examination, CDT: clock drawing test, T5P= Test delle 5 Parole.

**Figure 4 geriatrics-07-00049-f004:**
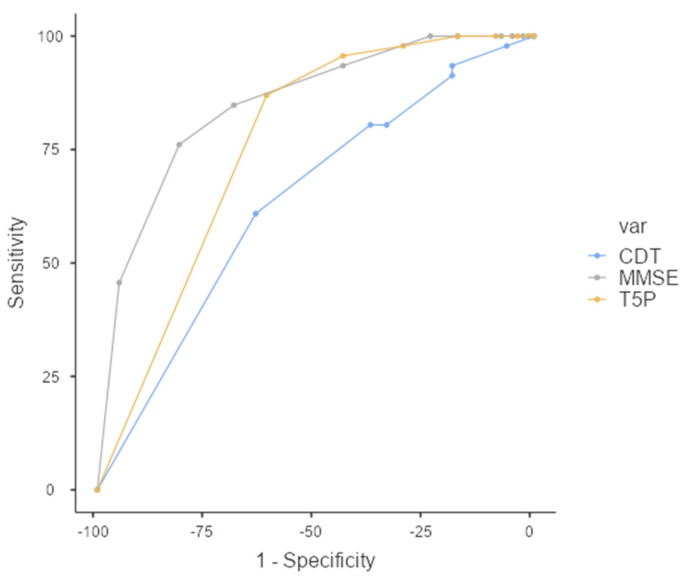
ROC Curve—T5P total score ≤ 9 discriminates between the MCI and FMD groups. MMSE: Mini-Mental State Examination, CDT: clock drawing test, T5P = Test delle 5 Parole.

**Table 1 geriatrics-07-00049-t001:** Demographic characteristics of the sample.

	AD (*n* = 60)	MCI (*n* = 80)	FMD (*n* = 46)	CTRL (*n* = 78)	Group Comparison *p*-Value (*)
**Age** (years; *mean ± SD*)	78.57 ± 4.443	76.18 ± 6.077	73.52 ± 5.320	76.03 ± 6.335	**<0.001**
**Education** (years; *Median*; IR)	8 (5–13)	8 (8–13)	8 (8–13)	8 (8–13)	0.180
**Sex** (M:F)	22:38	39:41	12:34	32:46	0.087

*n* = number, SD = Standard Deviation, M = Males, F = Female, IR = Interquartile Range, AD = Alzheimer’s disease, MCI = mild cognitive impairment, FMD = functional memory disorders, CTRL = control subjects.

**Table 2 geriatrics-07-00049-t002:** Sample size calculation performed with G power software.

**F Test**—ANCOVA: Fixed Effects, Main Effects And Interactions	
**Analysis:** Post Hoc: Compute Achieved Power—Given α, Sample Size and effect Sample Size	
**Input:** Effect size f	0.9192489
α err prob	0.001
Total sample size	264
Number of groups	4
Number of covariates	1
**Output:** Noncentrality parameter λ	223.0849
Critical F	3.0847131
Denominator df	259
**Power (1-β err prob)**	**0.999**

**Table 3 geriatrics-07-00049-t003:** Neuropsychological Assessment of the sample.

	AD (*n* = 60)	MCI (*n* = 80)	FMD (*n* = 46)	CTRL (*n* = 78)	Group Comparison *p*-Value (*)
**MMSE** (total score; *mean* *± SD*)	22.40 ± 2.901	26.53 ± 2.140	29.00 ± 1.229	28.83 ± 0.986	**<0.001**
**T5P** (total score; *mean* *± SD*)	6.23 ± 2.126	8.34 ± 1.814	9.80 ± 0.582	9.79 ± 0.466	**<0.001**
**CDT** (total score; *Median* *(IR)*)	5 (3–7)	7 (5–9)	9 (7–9)	9 (7–9)	**<0.001**

*n* = Number, SD = Standard Deviation, M = Males, F = Female, IR = Interquartile Range, AD = Alzheimer’s disease, MCI = mild cognitive impairment, FMD = functional memory disorders, CTRL = control subjects; MMSE = Mini-Mental State Examination; T5P = Test delle 5 Parole; CDT = clock drawing test.

**Table 4 geriatrics-07-00049-t004:** Correlation between T5P, CDT and MMSE scores.

	MMSE (Total Score)	CDT score	MMSE 3 Words Recall
**T5P** (total score)	**0.693** *	**0.368** *	0.565 *
**T5P Free Delayed Recall**	0.666 *	0.426 *	**0.557** *

(* significant different, *p* < 0.001). MMSE: Mini-Mental State Examination, CDT: clock drawing test, T5P = Test delle 5 Parole.

**Table 5 geriatrics-07-00049-t005:** ROC Curve—T5P total score ≤ 9.

Group Comparison	AUC	Sensitivity	Specificity
ADvsFMD	0.964 (outstanding)	86.96%	96.67%
ADvsCTRL	0.967 (outstanding)	82.05%	96.7%
MCIvsFMD	0.760 (acceptable)	86.96%	61.2%
MCIvsCTRL	0.752 (acceptable)	82.1%	61.25%

AUC = Area under the Curve, AD = Alzheimer’s disease, MCI = mild cognitive impairment, FMD = functional memory disorders, CTRL = control subjects.

## Data Availability

Data can be obtained upon reasonable request from the corresponding author.

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
