# Peer review of "Translation, Adaptation and Validation of the Five-Word Test (Test Delle 5 Parole, T5P) in an Italian Sample: A Rapid Screening for the Assessment of Memory Impairment"

_geriatrics, 2022, doi:10.3390/geriatrics7020049_

Round 1

Reviewer 1 Report

This is a possibly interesting work on an almost large sample of individuals. Nevertheless there are several issues that should be addressed before it could be considered for publication.

Introduction:

31-34 “Much more recently, the advent of new therapies (i.e., Aducanumab-avwa) that seem able to treat the neurodegenerative process raises the need to identify as early as possible, possibly in the prodromal phase, the presence of cognitive deficits related to Alzheimer's disease.”

This sentence is too strong and optimistic with respect to Adacanumab. Authors may mitigate suggesting that the identification of patients in the prodromal phase could be pivotal for inclusion in clinical trials, for instance.

41-43: “such as depressive and anxious disorders, sleep disorders, internist pathologies and the use of anticholinergic drugs and benzodiazepines”

Other neurologic diseases can cause memory disorders and are not mentioned by the authors, i.e. vascular dementia. This is also a limitation of the study, as authors did not consider other diseases causing dementia possibly mimicking AD.

45-46:  “the hippocampal amnesic syndrome typical of AD can be confused with memory dysfunction due to aging or depression”.

I don’t agree with authors. Memory impairment of AD is much more severe than in normal aging and could be hardly confused. Furthermore, MCI could also include patients with poor scores on cognitive tests due to normal aging and not due to AD. Therefore, it is not correct to distinguish between MCI and “memory dysfunction due to aging or depression”. Rather authors should focalize all the introduction and the aim of the study on distinguishing between patients with memory impairment possibly due to AD with patients with memory impairment due to non-AD pathology. However, this analysis would need biomarkers data, and this lack is a strong limitation of the study.

47-49: At this point, as before, patients only considered AD v.s. functional disorders, but FCRST could be used also to discriminate between AD and FTD, for instance.

Methods:

85-90: there are several concerns regarding recruitment of patients:

- Inclusion and exclusion criteria should be clearly exposed;

- Authors stated that clinical criteria were applied for diagnosis of MCI and AD. But how was cognitive decline assessed? Did they use only MMSE? This point is fundamental to exclude circularity (use of tests to be analyzed in the study as selection tools).

- How were patients with “functional memory disorders/depressive pseudodementia” selected? Why did authors consider them as a unique group? How did authors exclude other organic cognitive impairment?

- Pseudodementia refers to dementia due to depression in elderly. How was depression assessed?

Results:

Table 2: Patients with pseudodementia had higher scores compared to AD and MCI, with a mean score of 29. I think that this is the demonstration that authors confounded pseudodementia with subjective cognitive decline due to depressive syndrome. In the former case, there is a real cognitive decline (mainly characterized by attentive deficits) with poor scores on cognitive tests, while patients with SCD obtain normal scores on neuropsychological examination. A mean MMSE score of 29 is hardly consistent with pseudodementia if patients have not been assessed with a more extensive battery or considering other features (i.e. autonomy in daily life).

Correlations: Authors should test for correlation between neuropsychological and demographic variables.

163: “Total Score≤9” How was this cut-off identified? Only by visual evaluation of the ROC curve? If so, I suggest authors apply statistical methods (i.s. Youden index).

172: good sensitivity and specificity: looking at the table, the good specificity is quite questionable (61% is a low specificity)

177: this is not a demographic analysis. If authors want to report this data, they should perform a chi square test to assess if the differences in proportion are significant. But I think that this is needless.

180: “Interestingly, T5P Total Score≤9 can discriminate between PseD and MCI with better sensitivity than MMSE total score”. How do authors get to this conclusion? The sensitivity values are reported without confidence intervals, so authors are not able to state if the difference (86 vs. 84) is significant. As well they can not say that the specificity values are similar.

191: high sensitivity (56.25%). I suppose that authors are aware that 56% is not a high sensitivity.

192: In line with Dubois’ suggestions (personal communication to E.F.). What is the utility of this sentence? Why did the authors specify that this is a suggestion by Dubois?

Discussion

203-209: authors are very optimistic, in particular when they stated that “clinicians could treat the disease as early as possible, even before the onset of decline, thus enabling individuals to live long and productive lives free of dementia caused by AD”. Unfortunately, disease modifying therapies are not available, so we are not able to prevent dementia. This paragraph should be strongly mitigated.

232: “with high sensitivity and specificity” this is not true, as stated above.

 245: “this clinical condition is now considered as the prodromal phase of AD” this is not true: only MCI due to AD (MCI with b-amyloid pathology) is considered prodromal AD.

Line 252: “we also tested subjects with subjective memory disorders linked to depressive mood to evaluate the ability of T5P to the difference between MCI and Pseudodementia.” So were they patients with subjective memory disorders? Why did authors classify them as PseD?

Line 258: “Thus, AD amnesic syndrome can be sometimes confused with subjective memory dysfunction due to aging or depression”. How can AD (that present an objective impairment) be confused with subjective memory dysfunction, which are “subjective” by definition?

266-270: What is the utility of this paragraph? Authors received a suggestion by professor Dubois, but this suggestion did not lead to advantages in performance of the T5P. Again, why did the authors specify that this was a suggestion by Dubois?

273-275: “recommendable in routine clinical practice to screen persons with subjective/probable amnesic difficulties and discriminate between patients with AD or MCI and healthy controls (including persons with  subjective amnesic problems due to mood problems).” This is not a conclusion of the study. To get this conclusion, authors should have included only patients with “subjective/probable amnesic difficulties” and test if T5P could discriminate between patients with AD and patients without AD (i.e. by CSF biomarkers). But this is not the case. Authors simply showed that demented patients performed worse than non-demented patients on T5P. Nevertheless, to define a person as demented or non-demented a neuropsychological examination should be administered. Therefore authors are saying that a patients, who had a lower scores on neuropsychological test (as he is defined as demented), had lower scores on a memory test. So what is new?

277: “(due to the growth of the elderly)” this is a rather simplistic view, I suggest authors erase this sentence.

278-279: “this simple but effective tool could help general practitioners to discriminate between degenerative organic memory deficits and “functional” disturbances, playing a filtering and guiding role”

There are two main points against this conclusion: 1- not all the memory deficits are due to hippocampal memory impairment. For instance, authors did not consider vascular dementia, an organic condition that is characterized by memory disturbance but due to a different anatomo physiological substrate without hippocampal memory deficit. 2- The authors did not consider other neurodegenerative dementia.

Minor:

The sample size calculation should be reported as a table.

Figure 1 and figure 2: Numbers in the x-axis should be changed with group names.

165: “According to literature [12], an accuracy value between 0.8 and 0.9 can be considered an excellent value.” This sentence should be moved to the discussion

Reviewer 2 Report

  1. The introduction could be expanded. More info is needed about the difficulties of differential diagnosis between the three taxonomic categories (pseudo-dementia, AD, and MCI). The authors need to justify the need for such a psychometric tool. The authors could cite relevant studies to prove their point, i.e. that differentiating the two neurological entities (AD and MCI) is difficult in clinical practice, and therefore such a tool would have a high level of clinical utility.
  2. In the first paragraph of page two, the authors are discussing about the need for a way to discriminate between three diagnostic categories (PseD, AD, and MCI). However, when stating their aim, they argue this tool would be useful for the differential diagnosis between AD and MCI. This is confusing. If the tool can be used to discriminate between PseD, AD, and MCI, it should be clearly stated. If not, I do not see why PseD is mentioned in the introduction.
  3. I guess that Kolmogorov-Smirnov showed that education was not normally distributed, and that is why K-S was used to assess possible differences with regard to this variable. Please clarify this in the Statistical Analysis section.
  4. The output of the G-power analysis could be excluded from the manuscript.
  5. The x-axis in the error-bar charts is not clear. Please specify the groups (I’m guessing 1 is AD, 2 is MCI, 9 is PseD, and 10 is ctrls, but this has to be made clear).
  6. In the discussion (p.7, line 203), the authors claim that MCI is a prodromal phase of AD. This seems like an oversimplification. There are several subtypes of MCI (not only amnestic) and there is no conclusive evidence that MCI (regardless of the type or origin) will deterministically lead to AD. In this view, the first paragraph of the discussion should be revised.
  7. In relation to the above comment, in p 8, lines 244-5, the authors repeat their claim that MCΙ is a prodromal phase of AD. They cite Albert et al. (2011). However, Albert and colleagues do not support this view. Their paper reads:

“If an autosomal dominant form of AD is known to be present (i.e., mutation in APP, PS1, PS2), then the development of MCI is most likely the prodrome to AD dementia. The large majority of these cases develop early onset AD (i.e., onset below 65 years of age). There remains, however, variable certainty about the time course over which the progression from MCI to AD dementia will evolve in these individuals.”

Adopting this view, MCI could be the prodromal stage of AD under certain circumstances. Overall, the authors should revise the discussion in order to be more specific. I’m not sure whether the argument about an early diagnosis is truly valid. The authors do not include sufficient details in their Methods section (i.e., AD stage, genetic information about the MCI patients, etc.), in order for such a claim to be adequately supported.

  1. In the discussion, the authors state that they expected no efficacy of the T5P regarding discrimination between PseD and ctrls. But this expectation is not clearly presented in the introduction.

  1. The present results show similar discriminative utility of the T5P and the MMSE. It would help, if the authors elaborate on the possible advantages of T5P in clinical practice.
  2. It should be noted that T5P (as well as other tools, such as MMSE) is a screening tool that does not lead to a definitive diagnosis. A short clarification about the need for further cognitive assessment (after the administration of T5P) is essential for a diagnosis, would suffice.
  3. It seems to me that the authors discuss T5P, CDT and MMSE as if they were similar. However, the three tools are different kinds of tasks. Regardless of the fact that they are useful for detection of possible dementia, they assess different cognitive domains. This could be briefly discussed.

Round 2

Reviewer 1 Report

The manuscript was sensibly improved. I thank the authors for appreciating my suggestions

Author Response

We are very grateful to the Reviewer for all valuable suggestions, which have allowed us to increase the quality of our manuscript

Reviewer 2 Report

The authros have made substantial changes and the manuscript has been significantly improved. I only have two minor comments:

1. Why not use systematically “amnestic” instead of “mnestic”?

2. In the discussion (line 238), the paper reads: “AD and its prodromal phase (MCI; [21])”. I would suggest rephrasing: “AD and MCI, which could be a prodromal phase of AD”. The authors have done a great job elaborating in this paragraph about the possible course from MCI to AD.

Author Response

Thanks for you comments. As correctly suggested, we have: 

1. use systematically “amnestic” (instead of mnestic)

2. rephrase line 238 in the discussion: "This work fits the growing interest in early detection, and consequent timely treatment of AD and MCI, which could be a prodromal phase of AD [21]"

We are very grateful to the Reviewer for all observations, which have allowed us to increase the quality of our manuscript.